# Regulators at Every Step—How microRNAs Drive Tumor Cell Invasiveness and Metastasis

**DOI:** 10.3390/cancers12123709

**Published:** 2020-12-10

**Authors:** Tomasz M. Grzywa, Klaudia Klicka, Paweł K. Włodarski

**Affiliations:** 1Department of Methodology, Medical University of Warsaw, 02-091 Warsaw, Poland; tomasz.grzywa@wum.edu.pl (T.M.G.); klaudia.klicka@wum.edu.pl (K.K.); 2Doctoral School, Medical University of Warsaw, 02-091 Warsaw, Poland; 3Department of Immunology, Medical University of Warsaw, 02-097 Warsaw, Poland

**Keywords:** miRNA, tumor invasiveness, metastasis, cell migration, epithelial–mesenchymal transition, tumor suppressor miR, oncomiR

## Abstract

**Simple Summary:**

Tumor cell invasiveness and metastasis are key processes in cancer progression and are composed of many steps. All of them are regulated by multiple microRNAs that either promote or suppress tumor progression. Multiple studies demonstrated that microRNAs target the mRNAs of multiple genes involved in the regulation of cell motility, local invasion, and metastatic niche formation. Thus, microRNAs are promising biomarkers and therapeutic targets in oncology.

**Abstract:**

Tumor cell invasiveness and metastasis are the main causes of mortality in cancer. Tumor progression is composed of many steps, including primary tumor growth, local invasion, intravasation, survival in the circulation, pre-metastatic niche formation, and metastasis. All these steps are strictly controlled by microRNAs (miRNAs), small non-coding RNA that regulate gene expression at the post-transcriptional level. miRNAs can act as oncomiRs that promote tumor cell invasion and metastasis or as tumor suppressor miRNAs that inhibit tumor progression. These miRNAs regulate the actin cytoskeleton, the expression of extracellular matrix (ECM) receptors including integrins and ECM-remodeling enzymes comprising matrix metalloproteinases (MMPs), and regulate epithelial–mesenchymal transition (EMT), hence modulating cell migration and invasiveness. Moreover, miRNAs regulate angiogenesis, the formation of a pre-metastatic niche, and metastasis. Thus, miRNAs are biomarkers of metastases as well as promising targets of therapy. In this review, we comprehensively describe the role of various miRNAs in tumor cell migration, invasion, and metastasis.

## 1. Introduction

Tumor cell invasiveness is one of the hallmarks of cancer defined by Hanahan and Weinberg [1]. During oncogenesis, tumor cells acquire invasive potential, followed by the expansive growth and invasion of adjacent tissues and basement membrane. Tumor invasiveness is regarded as a heterogeneous and multistep process [2]. It is accompanied by angiogenesis, intravasation, and metastasis into the secondary site [3]. Initial steps in metastasis are completed very effectively, whereas the latest steps are very inefficient and are limiting for cancer progression [4]. Only 0.01% of cells that enter the circulation will successfully colonize distant sites [5]. Nonetheless, over 90% of the mortality due to cancer is attributable to metastases [6].

In 1993, Lee et al. described for the first time small RNA molecules encoded by the lin-4 gene regulating the expression of protein lin-14 in *Caenorhabditis elegans* [7]. Further studies revealed that microRNAs (miRNAs, miRs) are short single-stranded non-coding RNAs that regulate gene expression post-transcriptionally (Figure 1) [8]. Pri-miRNAs are transcribed in the nucleus by RNA polymerase II. Then, pri-miRNAs are cut by a protein complex consisting of Drosha and DGCR8. In the next step, pre-miRNA is exported to the cytoplasm and then cut by Dicer near the loop to form the miRNA duplex [9]. Cooperating with Argonaute proteins, miRNA creates an RNA-induced silencing complex (RISC) that targets mRNA and regulates genes the expression post-transcriptionally [10,11].

The mechanism of this regulation involves the direct silencing of mRNA by the inhibition of the translation or destabilization of mRNA achieved by a shortening poly(A) tail, 5′-to-3′ exonucleolytic decay, and decapping [11]. MiRNAs bind to complementary sequences in the 3′ untranslated region (UTR) of target mRNA [9]. It has been identified that over 60% of human protein-coding genes harbor conserved miRNA target sites [12]. By targeting multiple mRNAs, miRNAs are involved in the regulation of a wide range of cellular processes including cell proliferation, differentiation, and apoptosis. Thus, the dysregulation of miRNAs is involved in the pathogenesis of many diseases, including cancer [13].

MiRNAs may play opposite roles in cancer after being either oncomiRs or tumor suppressor miRs (Table 1) [14]. The complexity of their effects makes them key regulators of all hallmarks of cancer [15]. MiRNAs may affect (promote or suppress) cancer cell proliferation, genomic instability [16], and apoptosis [17]. Moreover, miRNAs regulate tumor cell metabolism [18], angiogenesis [19], and cancer immune escape [20]. MiRNAs may either regulate gene expression in the cell or can be released outside the cell leading to the regulation of gene expression in adjacent cells. Therefore, miRNAs are not only key regulators of cancer cells but also of the complex regulatory network of the tumor microenvironment [21,22,23].

MiRNAs expression is often aberrated in cancers and results in the dysregulation of gene expression. The pan-cancer analysis revealed a global downregulation of tumor suppressors by miRNAs in cancer cells [41]. Many oncogenes, including MYC, were reported to upregulate oncomiRs [9]. Moreover, oncogenes repress the expression of tumor suppressor miRNAs [42]. Mutations in proteins involved in miRNAs biogenesis and maturation, e.g., Drosha or Dicer, lead to the dysregulation of the expression of tumor suppressor miRs or oncomiRs resulting in cancer progression [9]. A study of Merritt et al. showed that the majority of ovarian tumor specimens had decreased Dicer or Drosha mRNA which affects miRNA expression [43]. Dicer is also downregulated by miRNAs, including miR-103/107 [44]. Moreover, epigenetic changes in miRNA promoters affect miRNA expression in cancer tissue [45].

MiRNAs have been reported to regulate every step of cancer progression. They promote or suppress primary tumor growth, local invasion, as well as metastasis (Figure 2). In this review, we aim to comprehensively review the role of miRNAs in the progression from a benign tumor to an invasive metastatic cancer.

## 2. Tumor Cell Migration and Local Invasion

Cell migration is necessary for the maintenance of homeostasis in the human body, enabling tissue repair, regeneration, and immune response. However, cell migration is also a crucial driver of cancer invasion and metastasis. It is known that these cells use multiple strategies for migration. Two main types of cells’ movement include single-cell migration and collective cell motility [6]. Single-cell migration is characterized by the lack of cell-to-cell interactions during migration and is divided into amoeboid-like, characteristic for leukemia or lymphoma cells, and mesenchymal migration, which occurs in stromal tumors or epithelial tumors after epithelial–mesenchymal transition (EMT) [2]. The multicellular migration includes collective cell migration [46] and collective cell invasion [47], which are characterized by the migration of a group of cells that retain cell-to-cell adhesions. In the collective cell migration model, the leader cell migrates according to the single-cell model and forms a proteolytic microtrack. It is excavated and expanded by the following multicellular group to form a larger path of migrating cells [2].

Cancer cell migration and local invasion are heterogeneous processes that include several steps regulated by diverse miRNAs. Multiple models have been established based on in vitro and in vivo studies [2]. In general, single-cell invasive migration consists of five molecular steps (1) the polarization of cytoskeleton by actin polymerization and the formation of pseudopod protrusion, (2) the recruitment and adhesion of cell surface receptors to extracellular matrix (ECM), (3) the focalized proteolysis of ECM, (4) the cell contraction by actomyosin, and (5) the rear-end retraction (Figure 3) [2].

### 2.1. Step 1: Polarization of the Cytoskeleton and Formation of the Leading Protrusion

Cell migration is initiated by the changes in the cell cytoskeleton that result in the formation of the protrusions of the cell membrane. These actin-based structures are termed filopodia, lamellipodia, invadopodia, and podosomes, based on their characters [48]. Despite being regulated mostly on posttranslational levels, multiple miRNAs regulate the reorganization and polarization of the cytoskeleton. The majority of miRNAs targeting the mRNA of cytoskeleton-regulating proteins are tumor suppressors and are downregulated in cancer. The overexpression of those miRNAs affects cytoskeleton remodeling and decreases cell migration and invasiveness.The first step of tumor cell invasion is the formation of the leading edge protrusion, which is controlled mainly by the members of the Rho family of small GTPases, Cdc42 and Rac [2]. Cdc42 is a direct target of miR-133 [49], miR-186 [50], miR-195 [51] and miR-330 [52]. Moreover, Cdc42 is targeted also by miR-137 which inhibits cell invasiveness [53]. Importantly, the expression of miR-137 gradually decreases during cancer progression due to epigenetic silencing at an early stage [54,55]. Cdc42 cooperates with Rac to promote cell invasiveness [56]. RAC1 is targeted by miR-142-3p and miR-145, leading to suppressed cell migration and invasiveness [57,58]. Moreover, miR-124 affects the subcellular localization of RAC1 [59]. MiRNAs regulate not only small GTPases but also their regulators, including guanine nucleotide exchange factors (RhoGEFs), GTPase-activating proteins (RhoGAPs), GDP dissociation inhibitors (RhoGDIs), and guanine nucleotide exchange modifiers (GEMs) [2,60].

Multiple miRNAs have been identified as crucial regulators of the actin cytoskeleton in cancer cells by direct targeting multiple cytoskeleton-associated proteins (Table 2). The reduced expression of miR-138, which targets RhoC and ROCK2, is associated with enhanced metastatic potential in oral squamous cell carcinomas [61]. The overexpression of miR-138 decreases tumor cell migration and invasiveness. Likewise, overexpression of miR-124 targeting ROCK1, a major downstream effector of RhoA and RhoC family members [62], results in the decreased length and number of actin fibers in cells as well as a reduction in long and thin protrusions on the cell surface [63]. Increased let-7b or miR-142-3p expression inhibits the formation of lamellipodia and filopodia which leads to the persistent stabilization of stress fibers [64,65]. MiR-145 promotes actin cytoskeleton rearrangements and cortical actin distribution, but it also reduces actin stress fiber and filopodia formation [66]. Thus, by targeting regulators of the actin cytoskeleton, miRNAs can potently affect cancer cell migration and invasiveness.

### 2.2. Step 2: Formation of Focalized Clusters by Recruitment and Adhesion of Cell Surface Receptors to ECM

The second step in the tumor invasion is the activation of signaling pathways that control the tumor cells cytoskeleton as well as the cell-to-cell and cell-to-matrix interactions [2]. Extracellular matrix (ECM) is a key component of the tumor microenvironment [137]. ECM constituents serve as co-receptors, ligands, and signal presenters. Mechanocoupling between the cytoskeleton and ECM is crucial for the initiation of cell migration. For instance, miR-25-3p decreases adhesion to collagens I, II, and IV as to fibronectin, laminin, and tenascin [138]. Cancer cells sense ECM components and their mechanical properties by multiple adhesion and signal-transducing receptors, including integrins, syndecans, or CD44.

#### 2.2.1. Integrins

The main mechanosensors and cell adhesion receptors for ECM are integrins, a family of 24 transmembrane proteins [139]. These heterodimeric αβ receptors bind either ECM proteins or membrane proteins on other cells. The activation of integrins by ligand binding leads to the formation of adhesome, which regulates multiple processes including cell proliferation, survival, differentiation, and migration [140]. Downstream integrin effectors include cytoskeletal adaptor proteins talin, paxillin, and kindlin as well as small GTPases Rac and Rho, that regulate cell protrusion and rear contraction [2]. Thus, the role of integrins goes far beyond only ECM–cell interaction. Moreover, the interaction between integrins and growth factor receptors regulates tumor growth and metastasis [141]. Importantly, the expression of a specific integrin determines the target organs for metastases [142]. Multiple studies reported disturbed integrins expression in cancer, thus, they are a promising target of therapy [143].

Integrin subunits are targeted by multiple miRNAs (Table 3). For instance, miR-31 directly targets integrins α2, α5, αV, and β3 leading to the inhibition of cell spreading in a ligand-dependent manner [144]. Moreover, miR-142-3p targeting integrin αV substantially decreases tumor cell invasiveness [65]. In addition to the direct targeting of integrin-coding mRNA, miRNAs may indirectly affect integrin levels. For example, miR-375 decreases HuD mRNA stability and translation and leads to a reduced expression of N-cadherin, RhoA, NCAM1, and integrin α1 [145].

#### 2.2.2. Podoplanin

Podoplanin is a transmembrane glycoprotein that mediates the degradation of ECM via controlling the stability of invadopodia [174]. Moreover, podoplanin binds the ERM (ezrin, radixin, moesin) protein family to enhance RhoA activity and cell invasiveness [175]. MiR-363 targets podoplanin, leading to the inhibition of cell migration and invasion, thus its level is downregulated in metastatic squamous cell carcinoma [176]. Similarly, podoplanin is also a target of miR-29b and miR-125a that are downregulated in glioblastoma [177].

#### 2.2.3. CD44

CD44 is a cell-surface glycoprotein that mediates cell–cell interactions and cell adhesion. CD44 binds to hyaluronic acid and with low affinity to heparan sulfate, fibronectin, and collagen [178,179] It is overexpressed in several cell types, including cancer stem cells [180]. The level of CD44 is strongly regulated by miR-34a, which is a key negative regulator of prostate cancer stem cells [181]. MiR-34a, via targeting CD44, also suppresses angiogenesis [182]. Moreover, CD44 is targeted by miR-199a in ovarian cancer cells and so it suppresses the invasiveness, tumorigenicity, and multidrug resistance [183]. On the other side, the interaction of the CD44 with hyaluronan promotes miR-21 expression leading to the increased expression of anti-apoptotic proteins Bcl-2 and inhibitors of the apoptosis family of proteins (IAPs) [184] as well as increased cell migration and invasiveness [185]. Similarly, CD44–hyaluronan interaction induces the expression of miR-10b, which upregulates RhoA and RhoC resulting in the cytoskeleton activation and increased tumor cell invasiveness [186].

#### 2.2.4. Syndecan-1

Another protein mediating cell–ECM adhesion is syndecan-1 (CD138). It is a heparan sulfate proteoglycan and one of the regulators of integrin function that is involved in the differentiation of tumor cells [187,188]. Syndecan-1 is targeted by miR-10b which promotes cancer cell motility and invasiveness [188].

#### 2.2.5. Focal Adhesion Kinase (FAK)

FAK is a crucial component of the focal adhesion complex and functions as an integrator to control cell migration. FAK transduces extracellular stimuli into intracellular signaling, inducing the reorganization of the cytoskeleton. FAK has been identified as a target of tumor-suppressor miR-7, which inhibits EMT and metastasis [189,190]. Similarly, miR-138 and miR-135 target FAK and inhibit tumor cells invasiveness [191].

#### 2.2.6. Production of ECM

MiRNAs are important modulators of major ECM components expression. MiR-200c targets fibronectin [192]. MiR-29c, which is downregulated in tumor cells, targets mRNA encoding ECM proteins, including collagens I, III, IV, and XV as well as laminin, and osteonectin [193]. Moreover, some miRNAs were reported to regulate collagen maturation, including miR-122 that targets prolyl 4-hydroxylase subunit alpha-1 (P4HA1) [194]. Two tumor suppressor miRNAs, miR-26 and miR-29 target lysyl oxidase-like 2 (LOXL-2), which is a collagen-modifying enzyme, crucial for tissue remodeling and metastasis [195].

#### 2.2.7. Cadherins

In addition to cell–ECM interactions, tumor cells have dysregulated the expression of cell-to-cell adhesion proteins, including cadherins. Changes in the expression of cadherins promote or inhibit cell migration and invasiveness. MiR-9 targets E-cadherin resulting in the activation of β-catenin signaling [24]. Inhibiting miR-9 leads to the inhibition of metastasis formation [24]. Similarly, miR-720 targets E-cadherin and promotes metastasis [196]. Conversely, miR-96 upregulates E-cadherin expression by direct binding, which leads to the enhanced cell-to-cell adhesion [197].

#### 2.2.8. JAM-A

Junctional adhesion molecule A (JAM-A) is a cell–cell adhesion protein and a key regulator of cell migration and invasion [198]. JAM-A has been identified as a direct target of miR-145, which finds its expression downregulated in cancer [66]. An increase in the miR-145 level leads to the inhibition of cell motility and invasiveness [66].

### 2.3. Step 3: Local Proteolysis of ECM

The proteolysis and remodeling of ECM are crucial for the invasiveness of the cancer cell. Moreover, ECM is a signal reservoir due to the binding of growth factors, sequestering them, and preventing their free diffusion [137]. The degradation of ECM releases growth factors, chemokines, and angiogenic factors, that promote tumor growth, invasiveness, and metastasis [199]. The main enzymes that generate paths for migrating cells are matrix metalloproteinases (MMPs) [200,201]. MMPs are a family of zinc-dependent endopeptidases that degrade all components of ECM. The formation of the invadopodia that promotes the degradation of ECM by the presentation of MMP-14 and secretion of MMP-2 and MMP-9 is a fundamental event during tumor cell invasion [202].

Multiple miRNAs have been identified as directly targeting MMP mRNA. These miRNAs are tumor suppressors and potently decrease tumor cell invasiveness. For instance, the overexpression of miR-874 targeting MMP-2 decreases tumor cells’ invasiveness in vitro as well as decreases tumor growth in vivo [39]. Moreover, downregulated in metastatic cancer miR-29c targets MMP-2 and integrin β1 [203]. The loss of miR-29c increases cell proliferation, adhesion to ECM, as well as migration, and invasiveness [203]. Importantly, miRNAs can also target components of transcription factors regulating MMP expression. Tumor suppressor miR-34a has been identified to target Fra-1 [204,205], a component of activator protein 1 (AP-1) necessary for MMP-1 expression [206]. On the other site, many oncomiRs target negative regulators of MMPs, increasing their expression and activity. A well described oncomiR-21 targets RECK, a membrane-anchored MMP inhibitor, and TIMP3, a tissue inhibitor of MMP activity [207]. The inhibition of miR-21 results in the downregulation of MMP activities and reduced the motility and invasiveness of tumor cells [207].

In addition to MMPs, disintegrin and metalloprotease domains with thrombospondins motifs (ADAMTSs) are important metalloproteases with a complex role in tumor biology [208]. Many of them were reported to be controlled by miRNAs. MiR-140, with a reduced expression in cancer, decreases ADAMTS5, and inhibits cell migration and invasiveness [209]. On the contrary, upregulated miR-365 promotes cell invasion by targeting ADAMTS1 [210]. MiR-221-3p targets ADAMTS6 [211]. Thus, miRNAs are crucial in the regulation of ECM proteolysis (Table 4).

### 2.4. Step 4: Cell Contraction by Actomyosin, Myosin II Activation by the Small GTPase Rho and Step 5: Rotation of the Adhesive Bonds on the Trailing Edge

Actomyosin is the primary source of force in mammalian cells. Actin filaments are highly plastic and change dynamically in the cell. Actin polymerization and depolymerization are regulated mainly by the Ras homologue (Rho) superfamily of small GTPases [252], that are involved in the control of cell cytoskeleton organization, thus cell motility [253]. MiR-21 targets tropomyosin 1, an actin-binding protein and a putative tumor suppressor [254]. The administration of the miR-21 inhibitor substantially decreases tumor growth [255]. The regulation of the latest stages of invasive cell migration that includes cell contraction and rear-end retraction is similar to the regulation of the first step (Figure 3, Table 1).

Taken together, miRNAs create a complex network of interactions to regulate cell invasiveness (Table 5). The loss of miRNAs suppressing invasiveness is a crucial step during oncogenesis that allows local invasion and metastasis. On the other site, the upregulation of miRNAs promoting invasiveness accelerates cancer progression by increasing cell motility, invasion, and metastasis.

## 3. Epithelial–Mesenchymal Transition (EMT)

Epithelial–mesenchymal transition (EMT) is a process that is crucial for embryogenesis, wound healing but also malignant progression. EMT leads to the changes in cell–cell and cell–ECM interactions, that allow the migration of epithelial cells and confer them to the mesenchymal phenotype [329]. The process can be reversed and it is called a mesenchymal–epithelial transition (MET) and is associated with the colonization of distant organs and the formation of metastases [330]. The most important steps are changes in the cell polarity, cytoskeleton and adhesion to other cells and the basement membrane (Figure 4). This allows cells to invade local structures and migrate to distant localizations. Moreover, by undergoing an EMT, carcinoma cells can acquire stem-like cell capabilities such as unlimited self-renewal [331]. EMT is characterized by the downregulation of E-cadherin and the upregulation of N-cadherin and proteases including matrix metalloproteinases such as MMP2 and MMP9 [332]. The most important transcription factors initiating EMT are SNAIL, TWIST, and ZEB. EMT transcription factors regulate the expression of multiple miRNAs [333]. One of the targets of ZEB1 is miR-34a, which regulates multiple properties of tumor cells, including cell migration and invasiveness [333].

All components of EMT signaling pathways are regulated by miRNAs post-transcriptionally [332] (Table 6). In general, EMT is promoted by oncomiRs and inhibited by tumor suppressor miRNAs (Figure 4). A group of oncromiRs called pro-EMT miRNAs promote EMT, tumor cell invasiveness, migration, and metastasis. For example, miR-9 and miR-92a bind to CDH1 which encodes E-cadherin [334]. MiR-10a promotes tumor cell migration and invasion by regulating EMT [335]. In cancer, the maturation of miR-10a is accelerated by XRN2, which leads to increased EMT and metastasis [336]. On the other side, the most important negative regulators of EMT are the miRNA-200 family members. MiR-200s target central regulators of EMT, ZEB1, and ZEB2 [337]. Similarly, miR-22 inhibits EMT via targeting EMT inducer SNAIL and ECM-remodeling MMP14, leading to the suppressed tumor growth, dissemination, and metastasis [231]. MiR-122 triggers a reverse process to EMT, mesenchymal–epithelial transition (MET), and causes cytoskeleton disruption, enhances cell adhesion, and inhibits the migration and invasiveness of cancer cells [80]. A similar effect was exerted by miR-200 family members that induced MET in cancer cell lines [338].The transformation of the growth factor-β (TGF-β) pathway is key signaling inducing EMT. Moreover, TGF-β controls other processes crucial for cancer progression including tumor cell proliferation and invasion. It was shown TGF-β regulates miRNAs expression but is also a target of miRNAs. Several miRNAs were described to be implicated in TGF-β-mediated EMT [339]. Among them, the miR-34 and miR-200 family seem to play the most important role as they form two negative feedback loops with transcription factors involved in EMT. MiR-34 participates with SNAIL1 with the first negative feedback loop and controls the initiation of the EMT process. TGF-β downregulates members of the miR-200 family by the methylation of their promoters and forms an autocrine TGF-β/miR-200b feedback loop. Thus, TGF- β induces EMT by miR-200/ZEB interaction [340]. TGF-β downregulates the expression of miR-584, a negative regulator of PHACTR1 (phosphatase and actin regulator 1), which in turn leads to actin rearrangement and cancer cell migration [81]. Moreover, TGF-β regulates miRNAs targeting adhesion genes [138].

Another crucial signaling pathway for tumor cell invasiveness and metastasis is Wnt/β-catenin. β-catenin-dependent canonical Wnt signaling regulates cell proliferation as well as the development and promotion of EMT, tumor cell invasiveness, and metastasis [341,342]. In the absence of Wnt, β-catenin forms a complex with the tumor suppressor adenomatous polyposis coli gene product (APC), glycogen synthase kinase 3 (GSK3), Axin, and casein kinase 1 (CK1) and is phosphorylated by CK1 and GSK3, which leads to the constant proteasomal degradation of β-catenin. After stimulation with the Wnt ligand, axin is recruited to the membrane complex of the Frizzled (Fz) receptor, low-density lipoprotein receptor-related protein 5 (LRP5) or LRP6 and the scaffolding protein Dishevelled (Dvl), which inhibits the phosphorylation of β-catenin, leading to its stabilization and accumulation in the nucleus [343]. Canonical Wnt signaling is regulated by multiple miRNAs. Wnt1 ligand is targeted by miR-122 [298], miR-148a [344], miR-148b [345], miR-152 [346], miR-329 [347]. Similarly, β-catenin is a target of multiple tumor suppressor miRNAs including miR-33a [348], miR-214 [349], miR-200c [350], miR-320a [351]. WWOX, a β-catenin inhibitor, is targeted by an oncogene, miR-153 [352]. The systemic administration of the miR-153 inhibitor suppressed the development of hepatocellular carcinoma in mice, while tumor cells with upregulated miR-153 expression exhibited increased growth [352]. The expression of miR-374a promotes cell migration and invasiveness by targeting crucial negative regulators of Wnt/β-catenin, including WIF1, PTEN, and WNT5A [353]. Moreover, other components of the Wnt signaling pathway are regulated by miRNAs, including Frizzled-7 by miR-27a [354], LRP6 by miR-202 [355] and miR-432 [356], axin 2 by miR-107 [357], and axin 2 and GSKβ by miR-1246 [358]. Altogether, miRNAs are important regulators of EMT and may either promote or suppress it by targeting different factors (Table 6).

## 4. Angiogenesis

### 4.1. Regulation of Angiogenesis by miRNA

Angiogenesis is a process of blood vessel creation. Tumor angiogenesis is classified as one of the hallmarks of cancer [410]. MiRNAs create a regulatory network that controls angiogenesis [411]. MiRNAs target multiple components of the angiogenesis regulatory pathway (Table 7). The tumor cell secretes miRNAs in exosomes to promote the angiogenesis of microvessel endothelial cells [412]. Moreover, multiple miRNAs either promote or inhibit endothelial cell proliferation, migration, and tube formation [408]. MiR-93 produced by glioblastoma cells promotes endothelial cell spreading, growth, migration, and tube formation, stimulating blood vessel formation and supporting tumor growth in vivo [173]. On the other hand, tumor suppressor miRNAs, like the miR-200 family suppress angiogenesis through multiple mechanisms, including targeting IL-8 and CXCL1. Moreover, tumor suppressor miRNAs inhibit crucial signaling pathways including PI3K/AKT, mTOR, and IGF1R pathways to inhibit angiogenesis. Targeting the miRNA-dependent regulation of angiogenesis seems to be a promising therapeutic approach [19]. Noteworthy, MMPs and the tissue inhibitors of metalloproteinases (TIMPs) are crucial for the ECM remodeling and angiogenesis [200], and their regulation by miRNAs was described above.

### 4.2. Vascular Endothelial Growth Factor-A (VEGF-A)

VEGF family members are growth factors that play a key role in angiogenesis. They bind to tyrosine kinase cell receptors, VEGFR-1, VEGFR-2, and VEGFR-3. Among them, VEGFR-2 is the most pro-angiogenic receptor [463]. Both, VEGF and VEGFR are regulated by miRNAs in cancer tissues. MiR-140-5p targets VEGF-A and suppresses angiogenesis and cell invasion [413]. Moreover, miR-205 and miR-497 bind directly to VEGF [414,415]. Expression of miR-140-5p, miR-205, and miR-497 targeting VEGF are substantially downregulated in cancer. VEGFR is targeted by miR-378a which acts as a suppressor of cell proliferation and invasion [425]. Furthermore, miR-497 via targeting VEGF suppresses cell proliferation and invasion and inhibits key signaling pathways as MEK/ERK and p38/MAPK [464].

### 4.3. Thrombospondin-1 (TSP-1)

TSP-1 is a known antioncogenic factor that controls many cellular processes. It influences cell proliferation, invasion, and migration. It functions as a ligand of CD47 [465]. MiR-467 binds to TSP-1 and promotes tumor angiogenesis and thus increases tumor growth. The expression of miR-467 is upregulated in tumor cells [428].

### 4.4. Platelet-Derived Growth Factor (PDGF)

PDGF, a member of the receptor tyrosine kinases family, is a growth factor involved in angiogenesis, cell proliferation, and migration. MiR-29a targets PDGFC and PDGFA and thus acts as a tumor suppressor [439]. Moreover, PDGF induces the expression of miRNAs, including miR-221, which regulates PDGF-induced EMT and cell migration [466].

### 4.5. Hypoxia-Inducible Factor 1 Alpha (HIF1a)

HIF-1α is a transcription factor that regulates angiogenesis, cell proliferation, and invasion. In normoxia, the proline residues of HIF-1α are hydroxylated, which is recognized by Hippel–Lindau tumor-suppressor protein (VHL) leading to the degradation in the proteasome [467]. HIF-1α accumulates in the cell under hypoxic conditions. Despite being regulated mainly by posttranslational modification, HIF-1α is also a target of several miRNAs, including miR-20b and miR-107 (Table 7).

VHL is described as a tumor suppressor gene and its inactivation may regulate cancer development and progression [467]. VHL is directly targeted by miR-21 in pancreatic cancer. The inhibition of miR-21 causes the suppression of tumor cell invasiveness via the HIF-1/VEGF pathway and the downregulating of MMP-2 and MMP-9 [445]. Moreover, miR-155 creates a signaling pathway with VHL/HIF/VEGF and regulates angiogenesis and the aggressive malignant phenotype of cancer cells [446].

## 5. Chemokines and Growth Factors

The mobilization of tumor cells from tissue-fixed state to migrating cells is regulated by multiple factors, including extracellular chemokines and growth factors. Several chemokines, CXCL12, CXCL10, CXCL11, CCL21, and CCL25 were identified as crucial in the induction of cell invasion [2,468]. CXCL12, which promotes invasiveness as well as the recruitment of monocytes to the tumor microenvironment, is a target of miR-342 [469]. The upregulation of miR-342 leads to the inhibition of tumor growth in vivo [469]. Moreover, the receptor for CXCL12, CXCR4, is targeted by miR-613, which suppresses cell invasiveness [470]. MiR-34a downregulates CXCL10 leading to a decrease in cell migration and invasiveness [471]. CCR7, a receptor for CCL21, promotes invasiveness and metastasis as well as regulates actin polymerization and pseudopodia formation is a target of Let-7a [472,473,474]. Moreover, CXCL11, which promotes cell migration, is a target of miR-144 [468,475,476]. Activation of the CXCL12/CXCR4 axis activates RhoA signaling, which regulates actin cytoskeleton and cell motility. This effect is mediated by the upregulation of lncRNA XIST, which acts as a sponge for miR-133a-3p, preventing RhoA downregulation, and promoting tumor cell invasion [76].

MiRNAs are important regulators of all crucial signaling pathways in cancer cells. They regulate the transduction of signaling from the growth factor receptors, including epidermal growth factor receptor (EGFR) [477], and regulate the MAPK signaling pathway [478,479], PI3K/Akt [480], p53 signaling [481], and JAK/STAT pathway [482]. For instance, SOCS2, a negative regulator of the JAK/STAT pathway, is targeted by multiple miRNAs including miR-196a, miR-196b, and miR-194 that promote cell migration, invasion, cell proliferation, and EMT [483,484].

## 6. Intravasation, Systemic Circulation, and Extravasation

After the local invasion, the tumor begins to grow. A fast increase in the cell number eventually leads to the dissemination of cancer cells to distant sites. Tumor dissemination occurs in early lesions as well as in mid- or late-stage tumors [485]. Cancer cells emigrate from the primary tumor to secondary sites via blood vessels, lymphatic vessels, interstitial fluid, and nerves [485,486,487]. Most miRNAs regulate multiple steps of metastasis, conferring tumor cells the ability to spread. For instance, miR-182, markedly overexpressed in metastatic cancer, targets four metastasis-suppressing genes [242]. The inhibition of miR-182 decreases cell migration and invasiveness as well as decreases the rate of tumor cells’ intravasation and metastasis to the lungs [242]. The first and critical step for metastasis is intravasation. To do this, tumor cells have to overcome the barrier of the basement membrane and the wall of the vessel.

### 6.1. Intravasation

Tumor cells secrete miR-105, which targets ZO-1, the tight junction protein-1 in endothelial cells. The exosome-mediated transfer of miR-105 from cancer cells destroys the integrity of endothelial monolayers, which enable intravasation [488]. Additionally, miR-181a disrupts the endothelial barrier by targeting Kruppel-like factor 6 (KL-F6), leading to the decreased expression of ZO-1, occluding, and claudin-5, which results in blood–tumor barrier permeability [489]. Tumor cells intravasation is also promoted by miR-21, which targets tumor suppressor Pdcd4 [29].

### 6.2. Systemic Circulation

Most of the tumor cells in the circulation are either killed by immune cells or die in the process called anoikis [490,491]. Natural killer (NK) cells are the main immune cells eliminating circulating tumor cells, thus suppressing metastasis. Circulating tumor cells use multiple mechanisms to escape from NK, including coating with platelets [492] and alterations in the expression of MHC molecules, NK cell ligand, and immune-checkpoints [490]. Importantly, miR-296-3p, which is overexpressed in metastatic cells, targets ICAM-1, rendering resistance to NK cells lysis in vasculature [491]. Moreover, Dicer-generated miR-222 and miR-339 suppress ICAM-1 on tumor cells, leading to the decreased susceptibility to cytotoxic T-cells cytolysis [490,491,493]. ICAM-1 is also a target of miR-296-3p, which enables invasiveness, intravasation, and enhances the survival of NK-resistant circulating tumor cells [491].

Anoikis is a form of programmed cell death induced by the loss of contact with the ECM or with other cells [494]. Anoikis depends on the activation of caspase and downstream pathways that includes the intrinsic and extrinsic apoptotic pathways [494]. Many miRNAs have been identified as crucial in the promotion or prevention of anoikis. MiR-141 enhances anoikis resistance and metastasis by targeting KLF12 [495]. Similarly, miR-214 promotes cell survival contributing to the enhanced metastasis of melanoma cells [32].

### 6.3. Extravasation

The extravasation of tumor cells determines their metastatic potential. Tumor cells were found to secrete extracellular vesicles (EVs) loaded with multiple miRNAs that are transferred to endothelial cells leading to changes in vascular permeability. Tumor-derived exosomes containing miR-181c are capable of destructing the blood–brain barrier by the dysregulation of the actin cytoskeleton via targeting PDPK1 [496]. Similarly, exosomal miR-25-3p promotes vascular permeability and angiogenesis by targeting KLF2 and KLF4, regulating tight junction proteins [497]. Moreover, miR-214 has been identified as crucial in the promotion of metastasis by an enhancement of extravasation [498]. On the other side, p38 activated by IL-1β promotes the expression of miR-31, which targets E-selectin [499]. This in turn leads to the decreased adhesion to the endothelium and inhibited transendothelial migration of tumor cells [499]. Similarly, tumor-suppressors miR-148b as well as miR-155 inhibit metastasis formation by affecting extravasation and survival [84,500].

## 7. Metastatic Colonization

The last stage of tumor invasion is the colonization of the secondary site. Metastasis to the sentinel lymph node is the most common and the most reliable factor for survival predicting in patients with different types of cancer [501]. Furthermore, tumor cells exit lymphatics, enabling systemic dissemination [502]. Tumor cells modulate premetastatic niches to enable the settlement and metastasis growth in tumor-draining lymph nodes or distant organs [5,142,501]. MiRNAs regulate this process either directly in tumor cells, promoting their migration, invasiveness, and survival or by affecting other cells in the premetastatic niches.

Tumor cells secrete multiple factors that reach distant sites via body fluids—blood, lymph, and interstitial fluid. The pro-metastatic secretome includes pro-angiogenic VEGF, PLGF, immunomodulating TGF-β, and S100 family proteins [142]. Moreover, tumors secrete extracellular vesicles that contain multiple proteins and miRNAs to prepare premetastatic niches [142,503]. It makes the tissue microenvironment supportive and receptive for the colonization by the metastatic tumor cell, according to the seed and soil hypothesis [504,505,506]. Premetastatic niche formation includes ECM remodeling, angiogenesis [507], and immune cell education towards a pro-metastatic phenotype [503]. All these processes are regulated by miRNAs in EVs. Secreted miRNAs, including miR-105-5p, miR-21-5p, miR-139-5p, regulate ECM remodeling by increasing the expression of MMPs in fibroblasts, as well as stimulate their proliferation creating a premetastatic niche [508]. Moreover, miR-122 in tumor-secreted extracellular vesicles reprograms the metabolism of stromal cells favoring a premetastatic microenvironment [509,510]. An exceptional pro-angiogenic and pro-metastatic role has been attributed to cancer stem cell (CSC)-released EVs containing miRNAs that regulate crucial biological processes [507,508].

Additionally, it seems that the secretion of tumor-suppressor miRNAs in the exosome is one of the mechanisms for decreasing their levels. Tumor cells secrete tumor-suppressors, including miR-23b, miR-224, and miR-921, which inhibit cell invasiveness, anoikis, angiogenesis, and metastasis [511]. Tumor suppressor miRNAs, miR-26, and miR-29, by targeting LOXL2, suppress tumor metastasis and the recruitment of myeloid cells to the metastatic site [195]. Moreover, miR-203 acts as a tumor suppressor miR and quells cancer cell proliferation and invasion [512]. However, miR-203 in exosome secreted by tumor cells promotes the polarization of monocytes into tumor-associated macrophages, thus supporting metastatic niche formation [512].

## 8. Tumor–Stroma Interactions

### 8.1. Cancer-Associated Fibroblasts (CAFs– Tumor Cells Interactions

Tumor cells and non-cancerous stromal cells interact with each other. Importantly, tumor invasiveness and metastasis greatly depend on stromal cells. Among the crucial stromal cells that induce ECM remodeling and enable cancer cell invasion are cancer-associated fibroblasts (CAFs) [513]. Cancer cells acquire migratory properties by the interaction between integrin α5β1 and fibronectin on the surface of CAFs, which enables migration through the ECM [514]. Tumor cells dysregulate miRNAs expression in resident fibroblasts favoring their polarization into tumor-promoting CAFs [515,516]. Many miRNAs have been identified as regulating CAF activation, including the miR-31, miR-214, miR-155 [515], and miR-200 family [517]. MiR-200s regulate collagen contraction by CAFs as well as trigger ECM remodeling, invasion, and tumor metastasis [517]. Similarly, miR-222 regulates CAFs’ reprogramming and its overexpression promotes fibroblast-induced cancer cell migration and invasiveness [518].

Moreover, also stromal cells secrete exosomes that regulate tumor cells. For instance, astrocytes secrete exosomes that contain miR-345 targeting KISS1, upregulate autophagy and promote brain invasion [519]. Additionally, astrocytes-derived exosomes contain miRNAs targeting PTEN, leading to its loss in brain metastasis [520].

Bones are a frequent location of solid tumors metastases. Tumor cells secrete factors that dysregulate miRNA expression in osteoclast, favoring bone metastasis and osteolysis [521]. Moreover, cancer-derived miRNA-218 decreases the production of type I collagen by directly targeting Col1a1 in preosteoblasts [522].

### 8.2. Immune Cells–Tumor Cells Interactions

MiRNAs are also important regulators of immune cells in the tumor microenvironment [22,523]. Tumor cells secrete miRNAs to directly suppress the antitumor response. The high expression of miR-424 in tumors decreases T-cell activation [524]. Similarly, a high level of miR-23a and miR-29a impairs the antitumor activity of cytotoxic T lymphocytes [525,526]. MiR-10b upregulated in tumor cells suppresses NK-mediated killing of tumor cells via targeting stress-induced cell surface molecule MICB [527].

In addition to directly suppressing antitumor immunity, miRNAs induce the polarization of immune cells [528]. Tumor cells as well as tumor-associated stromal cells secrete miRNAs that hijack immune cells to polarize them into immunosuppressive, tumor-promoting cells. MiRNAs may be secreted in extracellular vesicles, in the complexes with RNA-binding proteins including AGO2 and nucleophosmin, with lipoproteins, or by the direct exchange between cell via gap junctions [23,529]. In addition to directly suppressing antitumor immunity, miRNAs induce the polarization of immune cells [530]. Among others, cancer cells secrete miR-1246, which is delivered to macrophages and triggers the increased activity of TGF-β and an anti-inflammatory phenotype [530]. Similar effects are exerted by tumor-secreted miR-21 [531], miR-22-3p [532] and miR-203 [512]. Moreover, tumor-associated macrophages secrete miR-223 that promotes the invasiveness of tumor cells [533].

MiRNAs are also involved in the regulation of immune cell recruitment into the tumor microenvironment. MiR-155 enables the infiltration of innate immune cells and the suppression of antitumor immunity [534]. MiR-494 regulates the accumulation of myeloid-derived suppressor cells (MDSCs) and the inhibition of the miR-494 suppressed tumor growth and metastasis [535]. Moreover, miR-494 promotes arginase expression in MDSCs [535], which is crucial for the suppression of antitumor immunity [536]. Elevated TGF-β suppresses miR-34a which targets CCL22. Increased CCL22 production recruits regulatory T cells, which creates an immunosuppressive microenvironment and favors the colonization of tumor cells [537]. On the other hand, TGF-β promotes miR-155 and miR-21 expression in myeloid cells favoring polarization into immunosuppressive MDSCs [538].

## 9. miRNAs as Biomarkers in Cancer

Due to the dysregulated pattern of miRNA expression in cancer, miRNAs arose as promising biomarkers [539,540]. MiRNA profiling is feasible because of the stability of miRNAs and their presence in different body fluids, fresh frozen tissues, and even routinely collected formalin-fixed paraffin-embedded (FFPE) tumor tissue [541]. Multiple miRNAs have been identified as diagnostic or prognostic biomarkers [542,543]. Moreover, several miRNAs have been reported as an important prognostic marker of lymph node metastasis and distant organ metastasis (Table 8). MiR-21 which inhibits apoptosis [184] and potently promotes invasiveness (Table 5), correlates with the lymph node metastasis in many types of cancer [544], including breast cancer [545] and pancreatic ductal adenocarcinoma [546]. Similarly, miR-10b increases tumor cell migration, and invasiveness is a biomarker of distant metastasis in colorectal cancer [547]. Furthermore, as miRNAs regulate tumor cells’ response to the therapy [548], there are promising tools to monitor anticancer treatment.

## 10. Challenges for the Use of miRNAs in Clinical Oncology

MiRNAs seem to have the potential for therapeutic use [554,555,556]. However, the first clinical trials did not live up to expectations. The first trial tested the miR-34-based compound—MRX34—in several types of cancer. X34 is a liposomal miR-34a-mimic that entered the phase I study. MiR-34a is a tumor suppressor miRNA which targets several genes from the different oncogenic pathway. The results confirmed antitumor actions and showed acceptable safety when used twice a week in patients with different solid tumors in the advanced stadium [557]. However, further studies were terminated due to serious adverse events (NCT01829971, NCT02862145) [558]. Other compounds tested in clinical trials involved TargomiRs, targeted minicells containing miR-16 family (NCT02369198, NCT03713320) [559,560], and cobomarsen, an oligonucleotide inhibitor of miR-155 (NCT03837457, NCT02580552) [561].

Despite great expectations, only a few miRNA-based therapies were tested in clinical trials and did not achieve satisfactory effects. There are several crucial challenges for the use of miRNAs in oncology that limit their efficacy (Figure 5). The most important biological feature of miRNAs that leads to unpredictable results as well as putative multiple side-effects is the complexity of miRNAs–targets network. That is, numerous pathways are affected, hence any miRNA-based therapy will always have diverse effects depending on the initial expression of their targets. Moreover, miRNA off-targets on mismatched targets must be taken into account [562].

MiRNAs act as modulators of the levels of multiple rather than only strong post transcriptional repressors [563]. However, many targets of miRNAs remain unknown as most studies focus on simple miRNA–target axes. Importantly, the upregulation of a single miRNA affects the global regulation of gene expression by endogenous miRNAs [564]. Therefore, comprehensive studies on miRNA–mRNA interactions with the use of high-throughput methods are required.

Recent advances provided reliable models to investigate the complex role of miRNAs in cancer [565], since the role of miRNA depends on the context and may be modulated by the tumor microenvironment or therapy. Thus, miRNAs must be tested in in vivo preclinical studies in different models, since single miRNA may act as either oncomiR or tumor-suppressor miR, even in similar tumor types. Xenograft models of human tumor-derived cells in immune deficient mice are the most reliable to evaluate the in vivo potential of miRNAs as well as their therapeutic potential [565].

Importantly, regardless of their relatively high stability, unmodified miRNAs administered to the circulation are degraded quickly by serum RNases [566]. Thus, chemical modifications are required to increase miRNA stability, providing their longer half-life time and higher therapeutic efficacy [567,568]. Another major limitation of miRNAs therapy, common to almost all types of gene therapy, is the targeted delivery of oligonucleotides to cancer cells. Unmodified miRNAs poorly penetrate the cell membrane. Therefore, delivery vehicles are required. Currently, different types of vectors are being tested for miRNA delivery, including inorganic materials, lipid-based nanocarriers, cell-derived membrane vesicles and viral vectors [568].

miRNA delivery is limited by several factors including limited tumor penetration and unavoidable yet undesired delivery to healthy tissues, including immune cells and hepatocytes. Over 60% of lipid-conjugated miRNAs are accumulated in the liver [569]. This leads to multiple side effects, unpredictable results, interactions with other drugs, low therapy efficiency or a lack of therapeutic effects. Intratumor injections, targeted delivery, or adequate delivery routes may overcome these obstacles. The more targeted delivery of miRNAs to the tumor increases the amount of miRNA that reach destination cells [570]. A tumor targeting antibody-guided nanoparticles with miR-34a effectively reached the metastasis of melanoma, which increased the amount of absorbed miRNA in tumor cells [571,572]. A similar approach was tested in patient-derived xenografts of pancreatic ductal adenocarcinoma. Tumor-penetrating nanocomplexes targeting cell surface proteins carrying antimiR oligonucleotides inhibiting identified oncomiRs, which potently suppressed tumor growth [573].

The administration of high doses of miRNAs also increases a high risk of immune cell activation. Exogenous single-stranded RNAs and double-stranded duplexes are recognized by Toll-like receptors (TLRs), triggering the expression of pro-inflammatory cytokines, including interferons (IFNs) [574,575]. Indeed, the first-in-human clinical trial of miRNA therapy was closed early due to serious immune-related adverse effects that resulted in patients deaths [558].

Considering all the aforementioned limitations, despite a huge progress in our understanding of miRNA engagement in cancer, before successfully entering clinical medicine, more comprehensive studies are required. Not only on the mechanisms of miRNAs action, but also on the safety and specificity of miRNAs delivery.

## 11. Conclusions

In recent years, significant progress has been made in understanding the role of miRNAs in orchestrating cancer progression. Molecular studies enabled the identification of tumor suppressor genes and oncogenes as direct targets of miRNAs. Multiple reports described the role of miRNAs in promoting or suppressing tumor cell proliferation, migration, invasiveness, and metastasis. Moreover, miRNAs are important players in chemoresistance and tumor immune evasion. Importantly, the function of miRNAs is tissue specific as well as context dependent. Single miRNA may act as oncomiR that promotes tumor cell invasiveness and metastasis in one type of cancer but in another type of tumor it can act as a suppressor miR [14]. Some oncomiRs are specific biomarkers and their inhibition seems to be a promising therapeutic approach. However, due to fact that a single miRNA can target multiple mRNAs, further research and careful data analysis are necessary.

## Figures and Tables

**Figure 1 cancers-12-03709-f001:**
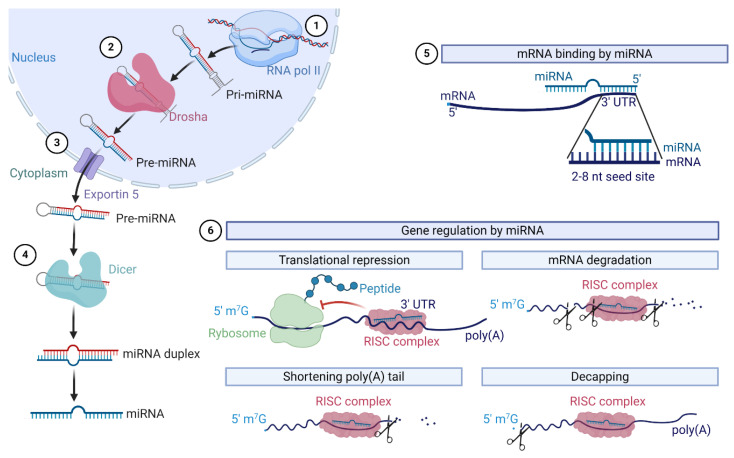
MiRNAs biogenesis and the mechanism of mRNA regulation. The crucial steps in microRNAs biogenesis include (**1**) transcription by RNA polimerase II; (**2**) the processing of pri-miRNA by ribonuclease Drosha; (**3**) transport into the cytoplasm by Exportin 5; and (**4**) the maturation of miRNA. The mechanism of miRNA action includes binding to the seed site of mRNA (**5**) and gene regulation by the RNA-induced silencing complex (RISC) complex (**6**) by translational repression, mRNA degradation, shortening poly(A) tail and the removal of 5′ 7-methylguanylate cap.

**Figure 2 cancers-12-03709-f002:**
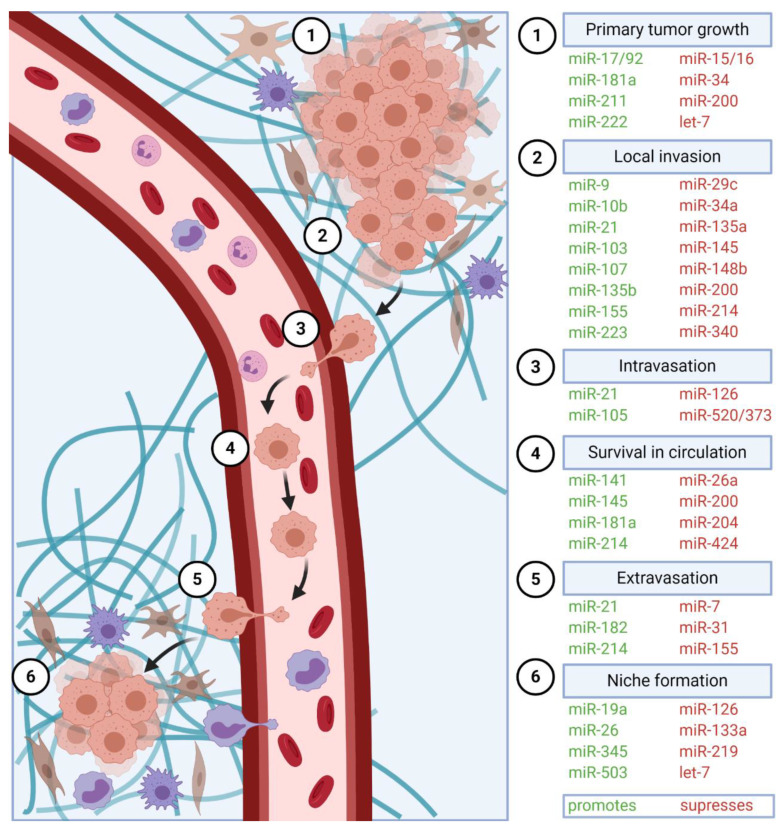
Regulation of cancer progression by miRNAs. Cancer progression involves several crucial steps, including (**1**) primary tumor growth, (**2**) migration and local invasion, (**3**) intravasation, (**4**) survival in the circulation, (**5**) extravasation, and (**6**) pre-metastatic niche formation (stromal cells, brown), recruitment of tumor-promoting immune cells (violet) and metastasis. Multiple miRNAs regulate each of these steps, and thus, act as either oncomiRs (promote cancer progression) or tumor suppressor miRs (suppress cancer progression).

**Figure 3 cancers-12-03709-f003:**
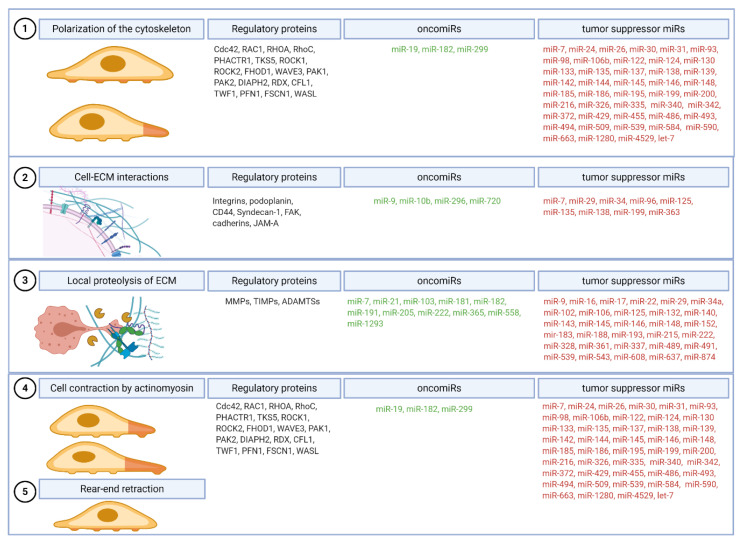
Stages of invasive cell migration. Five steps are required for successful tumor cell migration. The polarization of the cell cytoskeleton (**1**) begins the process of cell migration, followed by the formation of focalized clusters by the recruitment and adhesion of cell surface receptors to the extracellular matrix (ECM) (**2**) and the local proteolysis of ECM (**3**). Further steps include cell contraction by actomyosin (**4**) and the rotation of the adhesive bonds on the trailing edge (**5**). All steps are regulated by oncomiRs that promote each process and tumor suppressor miRs that inhibit cell migration and invasiveness by the regulation of mRNA of regulatory proteins.

**Figure 4 cancers-12-03709-f004:**
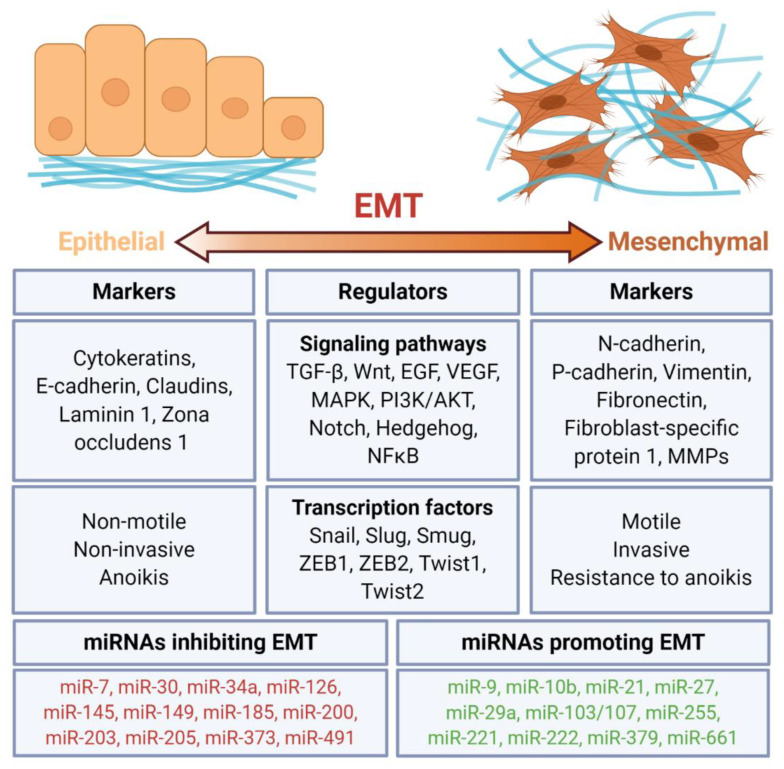
Regulation of epithelial–mesenchymal transition (EMT) by miRNAs. EMT is a process of the acquisition of the mesenchymal features, including motility, invasiveness, and resistance to anoikis of epithelial cells. It is regulated by multiple signaling pathways and transcription factors. Several miRNAs have been identified as either inhibiting or promoting EMT.

**Figure 5 cancers-12-03709-f005:**
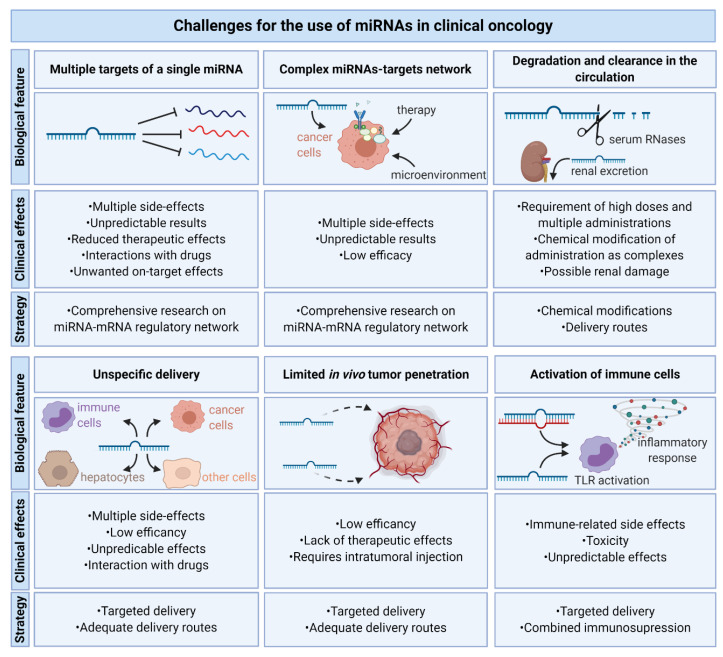
Challenges for the use of miRNAs in clinical oncology. Biological features of miRNAs include multiple targets, the complexity of the miRNAs–mRNAs network, the degradation by RNases and clearance in the circulation via renal excretion, unspecific delivery—not only to their destination, but also to healthy unaffected tissue including hepatocytes, limited in vivo tumor penetration and the activation of immune cells. This leads to a lack of therapy efficiency and there is a need for further comprehensive research on the miRNA–mRNA network and miRNAs’ delivery methods to overcome them.

**Table 1 cancers-12-03709-t001:** Role of miRNAs in cancer progression.

miRNA		Cancer Type	Target	Role in Vitro	Role in Vivo	Ref.
OncomiRs	miR-9	Breast cancer	CDH1	↑ cell migration and invasiveness	↑ tumor invasion and metastasis ↑ angiogenesis	[24]
	miR-10b	Breast cancer	HOXD10	↑ cell migration and invasiveness	↑ tumor invasion and metastasis	[25]
	miR-17-5p	Colorectal cancer	PTEN, P130	↑ cell migration and invasiveness	↑ tumor growth	[26,27]
	miR-19b	Breast cancer	TP53	↑ cell migration and invasiveness, cell cycle progression	↑ tumor growth and metastasis	[28]
	miR-21	Colorectal cancer	Pdcd4	↑ cell invasiveness	↑ intravasation and metastasis	[29]
	miR-135b	Lung cancer	LZTS1, Hippo pathway	↑ cell migration and invasiveness	↑ tumor growth and metastasis	[30]
	miR-181a	Breast cancer	Bim	↑ cell migration and invasiveness ↓ anoikis	↑ tumor growth and metastasis	[31]
	miR-214	Melanoma	TFAP2C, ITGA3	↑ cell migration and invasiveness	↑ extravasation and metastasis	[32]
	miR-211	Colorectal cancer	CHD5	↑ cells proliferation and migration	↑ tumor growth	[33]
	miR-223	Gastric cancer	EPB41L3	↑ cells motility and invasiveness	↑ metastasis	[34]
Tumor suppressor miRs	miR-34a	Neuroblastoma	MAP3K9	↑ induction of cell cycle arrest and apoptosis	↓ tumor growth	[35]
	miR-137	Colorectal cancer	FMNL2	↓ cells proliferation and invasion	↓ metastasis	[36]
	miR-192	Colon cancer	Bcl-2, ZEB2	↑ apoptosis	↓ metastasis	[37]
	miR-375	Liver cancer	AEG-1	↓ cell growth and invasiveness	↓ tumor growth	[38]
	miR-874	Non-small cell lung cancer	MMP-2, uPA	↓ cells invasiveness	↓ tumor growth	[39]
	let-7	Lung cancer	KRAS	↓ cell proliferation	↓ tumor growth	[40]

↑—increase, ↓—decrease.

**Table 2 cancers-12-03709-t002:** Direct regulation of actin cytoskeleton by miRNAs in cancer cells.

Target	Role	miRNA	Role	Ref.
RhoC	Promotes reorganization of the actin cytoskeleton and regulates cell shape and motility.	miR-93, miR-106b, miR-138, miR-372, miR-493, miR-509	Decreases migration and invasiveness, reorganization of the stress fibers.	[67,68,69,70,71,72,73]
CDC42	Transduces signals to the actin cytoskeleton, promotes the formation of filopodia.	miR-133, miR-137, miR-186, miR-195, miR-330	Decreases cell migration and invasiveness.	[49,50,51,52,53,74]
RAC1	Regulates reorganization of the actin cytoskeleton, promotes the formation of lamellipodia.	miR-142-3p, miR-145	Decreases cell migration and invasiveness.	[57,58]
RHOA	Regulates cell adhesion and migration, provides contractile force by the formation of stress fibers.	miR-31, miR-122, miR-133a-3p, miR-146a, miR-200b, miR-340-5p	Decreases cell migration.	[75,76,77,78,79,80]
PHACTR1	Binds actin and regulates the reorganization of the actin cytoskeleton.	miR-584	Decreases expression leads to the induction of migration.	[81]
TKS5	Regulates actin cytoskeleton and invadopodia formation.	miR-200c	Decreases invasiveness.	[82]
MYLK	Regulates the phosphorylation of myosin light chains.	miR-155, miR-200c	Decreases invasiveness and ability to form invadopodia.	[82,83]
ROCK1	Regulates actinomyosin cytoskeleton.	miR-124, miR-145, miR-148b, miR-199a, miR-335, miR-340, miR-584, miR-1280	Decreases cells migration and invasiveness.	[63,84,85,86,87,88,89]
ROCK2	Regulates actinomyosin cytoskeleton.	miR-124, miR-130a, miR-135a, miR-138, miR-139, miR-144, miR-185, miR-4529-5p	Decreases cells migration and invasiveness.	[90,91,92,93,94,95,96,97]
FHOD1	Regulates actin cytoskeleton.	miR-200c	Decreases migration and invasiveness.	[98]
PPM1F	Regulates actin cytoskeleton.	miR-149, miR-200c, miR-490, miR-590	Decreases migration and invasiveness.	[98,99,100,101]
WAVE3	Regulates actin cytoskeleton.	miR-31, miR-200	Decreases invasiveness.	[102,103]
PAK1	Regulates cell motility and cytoskeletal remodeling.	Let-7, miR-7, miR-26a, miR-26b, miR-98, miR-145, miR-485, miR-494,	Decreases migration and invasiveness.	[64,103,104,105,106,107,108,109]
DIAPH2	Regulates microtubule attachment to kinetochores.	Let-7, miR-10b	Decreases migration and invasiveness.	[64,110]
RDX	Regulates of membrane domains through interaction with the cytoskeleton and transmembrane proteins.	Let-7, miR-31, mir-200b, miR-409	Decreases migration and invasiveness.	[64,111,112]
PAK2	Regulates of cell motility and cytoskeletal remodeling.	miR-7, miR-23b, miR-137, miR-216a, miR-455, miR-4779,	Decreases migration and invasiveness, cytoskelet remodeling.	[113,114,115,116,117]
CFL1	Binds actin and regulates cell proliferation and migration.	miR-342	Decreases invasion and migration.	[118]
TWF1	Binds actin and promotes EMT.	miR-30c, miR-142, miR-486	Decreases stress fibers F-actin formation.	[119,120,121]
PFN1	Binds actin and inhibits cells proliferation, migration, invasion and EMT.	miR-19a-3p, miR-182, miR-299-3p, miR-330-3p,	Increases migration and invasiveness.	[122,123,124,125,126]
FSCN1	Stabilizes actin filaments in invadopodia.	miR-24, miR-133a, miR-133b, miR-145, miR-200b, miR-326, miR-429, miR-539, miR-663	Decreases invasiveness.	[127,128,129,130,131,132,133,134,135]
WASL	Regulates actin polymerization.	miR-142-3p, miR-148a	Decreases invasiveness, reduced number of membrane protrusions.	[65,136]

**Table 3 cancers-12-03709-t003:** The direct regulation of integrin subunits by miRNAs in cancer cells.

Integrin	Protein Name	Synonyms [146]	miRNA	Ref.
ITGA1	α1	CD49a	miR-375	[145]
ITGA2	α2	CD49b, α2 subunit of very late antigen 2 (VLA-2)	miR-31	[144]
ITGA2B	αIIb	GTA, CD41, GP2B, HPA3, CD41b, GPIIb	miR-122	[147]
ITGA3	α3	CD49c, α3 subunit of VLA-3	miR-199 family	[148]
ITGA4	α4	CD49d, α4 subunit of VLA-4	miR-30a	[149]
ITGA5	α5	CD49e, fibronectin receptor alpha	miR-25-3p, miR-26a, miR-31, miR-92, miR-142-3p, miR-148b, miR-149	[65,84,138,144,150]
ITGA6	α6	CD49f, ITGA6B	miR-25, miR-29s, miR-143-3p	[151,152,153]
ITGA7	α7		nd	
ITGA8	α8		nd	
ITGA9	α9		miR-296-3p	[154]
ITGA10	α10		miR-34a	[155]
ITGA11	α11		miR-148a	[136]
ITGAD	αD		nd	
ITGAE	αE	CD103, human mucosal lymphocyte antigen 1α	nd	
ITGAL	αL	CD11a (p180), lymphocyte function-associated antigen 1 (LFA-1) α subunit	miR-126	[156]
ITGAM	αM	Mac-1, CD11b, complement receptor 3 (CR3) subunit	miR-124 miR-223	[157,158]
ITGAV	αV	CD51, MSK8, vitronectin receptor α (VNRα)	miR-25, miR-31, miR-92, miR-142-3p	[65,144,151,159]
ITGAX	αX	CD11c, CR4 subunit	miR-142	[160]
ITGB1	β1	Fibronectin receptor β, CD29, MDF2, MSK12	miR-29b, miR-29c miR-31, miR-124 miR-130b, miR-149 miR-183, miR-338-3p, miR-451	[144,161,162,163,164,165,166,167]
ITGB2	β2	Leukocyte cell adhesion molecule, CD18, CR3 subunit, CR4 subunit	miR-1, miR-133a	[168]
ITGB3	β3	CD61; GP3A; GPIIIa, platelet glycoprotein IIIa	miR-31, mir-150	[144,169]
ITGB4	β4	CD104	miR-29a, miR-184	[170]
ITGB5	β5		miR-185	[171]
ITGB6	β6		miR-17/20a	[172]
ITGB7	β7		nd	
ITGB8	β8		let-7, miR-93 miR-145, miR-148a	[64,136,173]

nd—no data.

**Table 4 cancers-12-03709-t004:** Direct regulation of matrix metalloproteinases (MMPs) by miRNAs in cancer cells.

MMP	Role	Role in Cancer	miRNA	Ref.
**Collagenases**				
MMP-1	Degradation of collagen types I, II, III, V, IX and fibrillary collagen	Initial invasion, metastasis	miR-222, miR-361-5p	[212,213]
MMP-8	Degradation of collagen types I, II, III, V, IX and fibrillary collagen	Inhibits invasion and metastasis	miR-539, miR-2682-3p	[214]
MMP-13	Degradation of collagens types I, II, III, V, IX and fibrillary collagen	Tumor growth, invasion	miR-125b,miR-148a, miR-188-5p	[215,216]
**Matrilysins**				
MMP-7	Proteolysis of fibronectin, collagen type IV, laminin, elastin, entactin, osteopontin, and cartilage proteoglycan aggregates	Invasive potential, proliferation, anti-apoptotic	miR-143, miR-489, miR-543	[217,218]
MMP-26	Degradation of collagen type IV, fibronectin, fibrinogen, casein, vitronectin, and others	Activates MMP-9	nd	
**Metalloelastase**				
MMP-12	Degradation of elastin	Protective inhibition of tumor growth	nd	
**Stromelysins**				
MMP-3	Degradation of collagen types II, III, IV, IX, and X, proteoglycans, fibronectin, laminin, and elastin	Invasion, metastasis, EMT, angiogenesis	miR-17, miR-152	[219,220]
MMP-10	Degradation of proteoglycans and fibronectin	Invasion, migration, tumor growth	nd	
MMP-11	Degradation of alpha-1 antitrypsin	Early tumor invasion	miR-125a-5p, miR-145, miR-192	[221,222,223]
**Gelatinases**				
MMP-2	Degradation of type IV collagen	Degradation of ECM	miR-29b, miR-29c, miR-106b, miR-874	[39,203,224,225]
MMP-9	Degradation of type IV collagen	Degradation of ECM	miR-29b, miR-183, miR-491-5p	[226,227,228]
**Enamelysin**				
MMP-20	Tooth-specific MMP		nd	
**Membrane-Type**				
MMP-14	Degradation of fibronectin, collagen, and gelatin	Activation of other MMPs	miR-9, miR-22, miR-337-3p	[229,230,231]
MMP-15		EMT, angiogenesis	miR-608	[232]
MMP-16		Invasion, metastasis	miR-132, miR-146a, miR-146b, miR-215, miR-328-3p	[233,234,235,236,237]
MMP-17		Angiogenesis, metastasis	nd	
MMP-24		Migration, metastasis	nd	
MMP-25		Tumor growth	nd	
**Others**				
MMP-19		Tumor growth, adhesion, metastasis	miR-16, miR-193b-3p, miR-637	[238,239,240]
**Inhibitors of metalloproteinase**				
TIMP1	Inhibition of MMP-14 -16, -19, -24 and ADAM10	Inhibition of cancer growth and metastasis	miR-182, miR-1293	[241,242]
TIMP2	Inhibition of all MMPs and ADAM12	Inhibition of cancer growth and metastasis	miR-205-5p	[243]
TIMP3	Inhibition of all MMPs and ADAM10, 12, 17, 28 and 33; ADAMTS-1, -4, and -5, ADAMTS-2	Inhibition of tumor growth, angiogenesis, and invasion	miR-21, miR-103, miR-181b, miR-191	[244,245,246,247]
TIMP4	Inhibition of most MMPs and ADAM17d, -28, and -33	Inhibition of angiogenesis, and invasion, promotion of apoptosis	miR-558	[248]
RECK	Inhibition of MMP-9	Inhibition of metastasis	miR-7, miR-21, miR-222	[207,249,250,251]

nd—no data.

**Table 5 cancers-12-03709-t005:** MiRNAs regulating cancer cell invasiveness and their direct targets.

miRNAs Promoting Invasiveness			miRNAs Suppressing Invasiveness		
miRNA	Targets	Ref.	miRNA	Targets	Ref.
miR-9	SOX7, CDH1, α-catenin	[24,256,257,258]	miR-10b	IGF-1R, HOXA-3, FGF13	[259,260,261]
miR-10b	TP53, PAX6, NOTCH1, HOXD10, TIP30, KLF4, HOXB3	[259,262,263,264,265]	miR-29c	CDK6, ITGB1, TIAM1, Collagens, Laminin γ1	[193,266,267,268]
miR-21	PDCD4, maspin, HBP1, LZTFL1, KLF5	[185,269,270,271,272]	miR-34a	SATB2, BCL-2, HNF4α, Snail, MMP9, MMP14, Notch1	[273,274,275,276,277,278]
miR-103	DAPK, KLF4, OLFM4, LATS2, PTEN	[279,280,281,282]	miR-135a	ROCK1, Smo, ERRα	[283,284,285]
miR-107	TPM1, DAPK, KLF4	[279,286]	miR-145	PAK4, ROCK1, MMP11, Rab27a, MUC1, MMP16, N-cadherin, ZEB2, Ets1, KLF4	[88,287,288,289,290,291,292,293]
miR-135b	NR3C2, LZTS1, APC, FOXO1, ST6GALNAC2, RECK, EVI5	[30,71,294,295,296,297]	miR-148b	WNT1, MTA2, ROCK1, Dock6	[298,299,300,301]
miR-155	DOCK-1, SDCBP, ANXA-2, CLDN-1, NDFIP1, SOCS1, TP53INP1, BCL6	[302,303,304,305,306]	miR-200	Foxf2, Flt1, BMP4, Onecut2, LIMK1, BMI-1, E2F3	[307,308,309,310,311]
miR-223	PAX6, hFBXW7, EPB41L3	[34,312,313]	miR-214	JAG1, ROCK1, CDC25B, ARL2, GALNT7, MKK3, JAK1	[314,315,316,317,318,319,320]
miR-424	CADM1, SMAD7	[321,322]	miR-340	NT5E, EphA3, SIRT7, NF-κB1, RhoA, ROCK1, JAK1	[85,323,324,325,326,327,328]

**Table 6 cancers-12-03709-t006:** Direct regulation of EMT signaling by miRNAs.

Target	miRNAs	Ref.
TGF-β1	miR-99a, miR-99b, miR-744	[359,360]
TGFBR2	miR-17 family, miR-21, miR-204, miR-211, miR-373, miR-520	[361,362,363,364,365]
ZEB1	miR-200 family, miR-205	[366,367]
ZEB2	miR-132, miR-138, miR-154, miR-200 family, miR-205, miR-215	[367,368,369,370,371]
Twist1	let-7d, miR-33a, miR-145a-5p, miR-151-5p, miR-214, miR-580	[372,373,374,375]
Twist2	miR-138, miR-215	[376,377]
Notch	miR-23b, miR-30a, miR-34a, miR-206	[131,378,379,380]
Snail1	miR-22, miR-34a, miR-137, miR-182	[231,276,381]
Snail2	miR-30a, miR-124, miR-203, miR-204, miR-211	[362,382,383,384]
EZH2	miR-138	[368]
Slug	miR-1, miR-30a, miR-124, miR-506, miR-630, miR-590-3p	[385,386,387,388,389,390]
N-cadherin	miR-145, miR-194	[391,392]
Vimentin	miR-30c	[393]
Fibronectin	miR-1, miR-139, miR-200c, miR-432	[394,395,396,397]
E-Cadherin	miR-10b, miR-22, miR-23b, miR-25, miR-92a, miR-221, miR-720,	[196,398,399,400,401,402,403]
ZO-1	miR-103	[404]
Claudins	miR-98 (claudin-1), miR-146-5p (claudin-12), miR-421 (claudin-11), miR-488 (claudin-2), miR-155 (claudin-1)	[405,406,407,408,409]

**Table 7 cancers-12-03709-t007:** The direct regulation of the angiogenesis pathway by miRNA.

Target	miRNA	Ref.
VEGF	miR-20b, miR-27b, miR-29b, miR-93, miR-126, miR-128, miR-140-5p, miR-195, miR-203, miR-205, miR-497, miR-503, miR-638,	[51,245,413,414,415,416,417,418,419,420,421,422,423,424]
VEGFR	miR-378a, miR-497	[425]
NRP1	miR-141, miR-338	[426,427]
TSP-1	miR-19a, miR-182, miR-467	[428,429,430]
FGF	miR-503, miR-5582-5p	[421,431]
HDGF	miR-139, miR-195, miR-214, miR-497, miR-511, miR-873, miR-939	[432,433,434,435,436,437,438]
Angiogenin	miR-204	[363]
PDGF	miR-29a	[439]
HIF1a	miR-20b, miR-33a, miR-107, miR-135b, miR-519c	[440,441,442,443]
HIF2a	miR-145	[58]
VHL	miR-21, miR-155, miR-222	[444,445,446]
STAT3	miR-125a, miR-411, miR-544, miR-874, miR-1299	[447,448,449,450,451]
Bmi-1	miR-16, miR-132, miR-183, miR-200c, miR-203, miR-218	[452,453,454,455,456,457]
E2F3	miR-194-5p, miR-200c, miR-217, miR-432, miR-449a	[311,458,459,460,461]
NF90	miR-590-5p	[462]

**Table 8 cancers-12-03709-t008:** MiRNAs as biomarkers of metastasis.

miRNA	Cancer	Type of Tissue	Mirna Level	Lymph Node Metastasis	Distant Metastasis	Ref.
miR-203	Colorectal	Serum	High	OR 2.9; 95%CI 1.4–6.1; *p* = 0.0035	OR 5.3; 95%CI 2.4–11.5; *p* < 0.0001	[549]
miR-885-5p	Colorectal	Serum	High	OR 3.0; 95%CI 1.3–7.2; *p* = 0.0116	OR 3.1; 95%CI 1.0–10.0; *p =* 0.0456	[547]
miR-19a	Various carcinomas	Serum and tissue	High	OR 0.564; 95%CI 0.346–0.921	nd	[550]
miR-20a	Cervical	Serum	High	OR 1.552; 95%CI 1.137–2.118	nd	[551]
miR-21	Breast	Serum and tissue	High	OR 2.36; 95%CI 1.04–4.78; *p* = 0.03	nd	[545]
miR-21	Pancreatic ductal adenocarcinoma	Serum and tissue	High	OR 1.45; 95%CI 1.02–2.06; *p* = 0.038		[546]
miR-122-5p	Colorectal	Serum	High	OR 1.621; 95%CI 1.255–2.092; *p* = 0.0002	nd	[552]
miR-146b-5p	Colorectal	Serum	High	OR 2.096; 95%CI 1.594–2.756; *p* < 0.0001	nd	[552]
miR-186-5	Colorectal	Serum	High	OR 2.910; 95%CI 1.810–4.678; *p* < 0.0001	nd	[552]
miR-193a-5p	Colorectal	Serum	High	OR 0.656; 95%CI 0.577–0.774; *p* < 0.0001	nd	[552]
let-7i	Colorectal	Tissue	Low	nd	OR 5.5; 95%CI 1.1–26.8; *p* = 0.0334	[547]
miR-10b	Colorectal	Tissue	High	nd	OR 4.9; 95%CI 1.2–19.7; *p* = 0.0248	[547]
miR-29a	Colorectal	Serum	High	nd	OR 3.500; 95%CI 1.274–9.617; *p* < 0.05	[553]

OR—odds ratio, 95%CI—95% confidence interval, nd—no data.

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
