# Peer review of "Regulators at Every Step—How microRNAs Drive Tumor Cell Invasiveness and Metastasis"

_cancers, 2020, doi:10.3390/cancers12123709_

Round 1

Reviewer 1 Report

This review presents in a very didactic way the role of miRNAs in tumour progression, particularly in the metastatic process. This review is very complete with more than 550 references. Some additional diagrams would be welcome, in particular summarising the mode of action of miRNAs (third paragraph of the introduction).

Author Response

We thank Reviewer for the appreciation our work. We have added two diagrams, according to the Reviewer’s suggestion. Figure 1 presents miRNAs biogenesis and mechanism of mRNA regulation and Figure 5 presents challenges for the use of miRNAs in clinical oncology

Reviewer 2 Report

Strength:

The review provides a very detailed description and dissection of the molecular mechanisms of the various microRNAs regulating the metastasis mechanisms of the neoplasia. The different microRNAs involved in this process are analyzed step by step in a logical way. It identifies all the relevant microRNAs dividing them between pro-metastatic and antimetastatic. The review critically analyzes how these mechanisms influence in the metastatic process. The review brings all these relevant studies together and can serve as a reference material for the involvement of microRNAs in pro-and anti-tumor metastatic process. The great advantage of the work is the order and clarity in the presentation of state-of-art about microRNAs.

Limitations:

The review seems to be an accumulation of facts and data. The take home message is that the mechanisms are very complex. There is not a guide how one might use microRNAs to tilt the balance of these sometimes-opposing mechanisms in favor of anti-tumor effect. No major hypotheses were derived, and suggestions made as to how to test them. One major limitation in the development of microRNAs in the therapy of cancer is the lack of translatable models to test ideas and molecules. There is no discussion of what type of models are available to guide clinical studies. Finally, this work does not add new data or provide new hypotheses.

Conclusion:

In conclusion, the review includes a balanced, comprehensive, and critical view of the research area. I think it is a clear and well-illustrated review of the role of microRNAs in the metastatic process. Although the article lacks originality, I think this review is a good starting point for the reader and researcher interested in this topic.

Author Response

Reviewer #2

Strength:

The review provides a very detailed description and dissection of the molecular mechanisms of the various microRNAs regulating the metastasis mechanisms of the neoplasia. The different microRNAs involved in this process are analyzed step by step in a logical way. It identifies all the relevant microRNAs dividing them between pro-metastatic and antimetastatic. The review critically analyzes how these mechanisms influence in the metastatic process. The review brings all these relevant studies together and can serve as a reference material for the involvement of microRNAs in pro-and anti-tumor metastatic process. The great advantage of the work is the order and clarity in the presentation of state-of-art about microRNAs.

Authors’ reply:

We thank Reviewer for the appreciation our work.

Limitations:

The review seems to be an accumulation of facts and data. The take home message is that the mechanisms are very complex. There is not a guide how one might use microRNAs to tilt the balance of these sometimes-opposing mechanisms in favor of anti-tumor effect. No major hypotheses were derived, and suggestions made as to how to test them. One major limitation in the development of microRNAs in the therapy of cancer is the lack of translatable models to test ideas and molecules. There is no discussion of what type of models are available to guide clinical studies. Finally, this work does not add new data or provide new hypotheses.

Authors’ reply:

We have modified our manuscript in response to Reviewer’ suggestion. We discussed major limitations of the translational relevance of miRNAs studies as well as challenges for the use of miRNAs in clinical oncology. We have added two figures, including figure 5 depicting most important obstacles for the miRNAs in clinical oncology. We suggested that miRNAs are promising biomarkers in cancer, however, more research are required to determine whether miRNA-based therapies can be successful in humans.

Moreover, we highlighted the role of global, high-throughput methods in the studies on the complex role of miRNAs in cancer and in vivo studies using patient-derived xenografts as a gold-standard in miRNAs studies.

Conclusion:

In conclusion, the review includes a balanced, comprehensive, and critical view of the research area. I think it is a clear and well-illustrated review of the role of microRNAs in the metastatic process. Although the article lacks originality, I think this review is a good starting point for the reader and researcher interested in this topic.

Authors’ reply:

We thank Reviewer for the comments that improved our manuscript.
